# Alleviating Hallucinations in Large Vision-Language Models through Hallucination-Induced Optimization

**Xinyu Lyu[1,5]**[*]
xinyulyu68@gmail.com

**Beitao Chen[2]**[*]
chenbeitao@gmail.com

**Lianli Gao[2]**[†]
lianli.gao@uestc.edu.cn

**Jingkuan Song[2]**
jingkuan.song@gmail.com

**Heng Tao Shen[3,4]**
shenhengtao@hotmail.com

[1]Southwestern University of Finance and Economics, Chengdu, China
[2] Shenzhen Institute for Advanced Study,
University of Electronic Science and Technology of China
[3]Center for Future Media, University of Electronic Science and Technology of China
[4]Tongji University
[5]Engineering Research Center of Intelligent Finance, Ministry of Education
https://github.com/BT-C/HIO

## Abstract

Although Large Visual Language Models (LVLMs) have demonstrated exceptional abilities in understanding multimodal data, they invariably suffer from hallucinations, leading to a disconnection between the generated text and the corresponding images. Almost all current visual contrastive decoding methods attempt to mitigate these hallucinations by introducing visual uncertainty information that appropriately widens the contrastive logits gap between hallucinatory and targeted ones. However, due to uncontrollable nature of the global visual uncertainty, they struggle to precisely induce the hallucinatory tokens, which severely limits their effectiveness in mitigating hallucinations and may even lead to the generation of undesired hallucinations. To tackle this issue, we conducted the theoretical analysis to promote the effectiveness of contrast decoding. Building on this insight, we introduce a novel optimization strategy named Hallucination-Induced Optimization (HIO). This strategy seeks to amplify the contrast between hallucinatory and targeted tokens relying on a fine-tuned theoretical preference model (i.e., Contrary Bradley-Terry Model), thereby facilitating efficient contrast decoding to alleviate hallucinations in LVLMs. Extensive experimental research demonstrates that our HIO strategy can effectively reduce hallucinations in LVLMs, outperforming state-of-the-art methods across various benchmarks. Code is released at `https://github.com/BT-C/HIO`.

## 1 Introduction

The recent success of Large Vision-Language Models (LVLMs) marks a major milestone in artificial intelligence research [OpenAI, 2023, Alayrac et al., 2022, Li et al., 2023a, Liu et al., 2023c, Zhu et al., 2023, Bai et al., 2023, Dai et al., 2023, Wang et al., 2023b, Driess et al., 2023]. By seamlessly integrating visual cues with Large Language Models (LLMs), LVLMs have demonstrated unparalleled expertise in multimodal comprehension, logical reasoning, and interactive engagement. This

---

[*]Equal contribution.
[†]Corresponding author.

38th Conference on Neural Information Processing Systems (NeurIPS 2024).

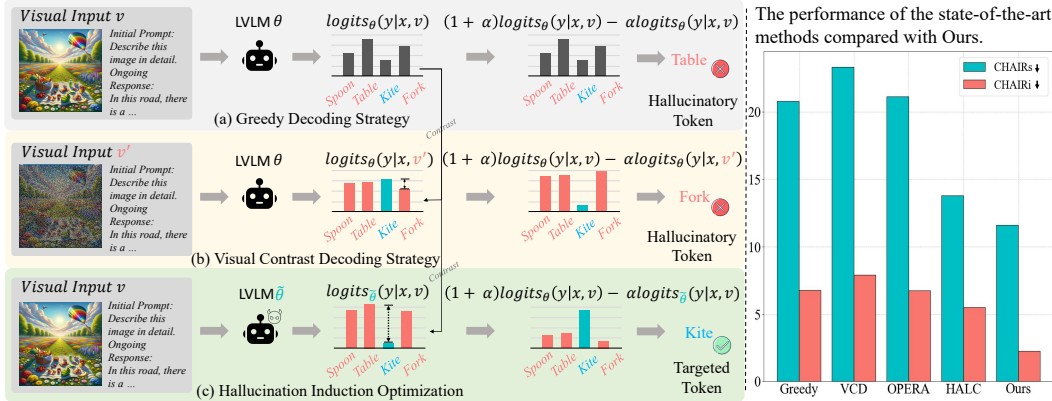

Figure 1: **(Left) Challenges and Solutions of Contrast Decoding Strategy.** Visual Contrastive Decoding, despite introducing perturbations to induce hallucinations, fails to effectively enlarge the logits gap between hallucinatory and targeted tokens, resulting in unsatisfactory outputs. On the contrary, our method addresses the issue by significantly amplifying the logits gap between hallucinatory and targeted tokens. **(Right) The performance of various methods on CHAIR metrics.** Our HIO generates descriptions with fewer hallucination tokens compared to other visual contrastive decoding methods, achieving lower scores on the CHAIRs and CHAIRi metrics.

integration has ushered in a new era in AI, breaking through traditional limitations and enabling a more holistic understanding of complex information OpenAI [2023], Yang et al. [2023], Lu et al. [2023], Yuan et al. [2022], Sun et al. [2024]. Despite these advancements, certain challenges remain, particularly the issue of hallucination Li et al. [2023b], Gunjal et al. [2023], Liu et al. [2023b], Lovenia et al. [2023]. Hallucination occurs when the language model generates content that deviates from the image's actual content, including imagined objects, fabricated scenes, incorrect spatial relationships, and misidentified categories.

Substantial research efforts have been directed towards mitigating hallucinations in Large Vision-Language Models (LVLMs). These efforts include post-hoc correction methods that refine LVLM outputs after the fact Zhou et al. [2023] and self-correcting frameworks specifically designed to reduce object hallucinations Yin et al. [2023]. Additionally, numerous decoding strategies have been developed to minimize hallucinations through the enhanced use of textual and visual priors Leng et al. [2023], Zhang et al. [2024], Favero et al. [2024], Zhu et al. [2024], Wang et al. [2024], Chen et al. [2024]. These methods aim to alleviate hallucinatory tendencies by integrating visual uncertainty, thereby increasing the contrastive disparity between hallucinatory and target logits. For example, Leng et al. [2023] augment the hallucinatory effect by introducing Gaussian noise into the images. Similar approaches by Zhang et al. [2024] and Favero et al. [2024] introduce substantial image noise, effectively reducing the original image to pure noise or unrecognizable content. Zhu et al. [2024] use instructional bias to enable the model to amplify its own hallucinations, while Wang et al. [2024] focus on deliberately amplifying the inherent image bias in LVLMs.

However, the inherent uncontrollable nature of global visual uncertainty challenges the precise induction of hallucinatory tokens. This limitation significantly undermines the effectiveness of these methods in reducing hallucinations and may inadvertently lead to undesired hallucinatory outputs. As shown in the left portion of the Fig. 1 *Spoon*, *Table*, and *Fork* are identified as hallucinated words, while *People* being the accurate term. For Greedy Decoding method shown in Fig. 1 (a), *Table* is selected as the final output based on the logits distribution. Moreover, although Visual Contrastive Decoding introduces perturbations to images to enhance hallucinations in Fig. 1 (b), it fails to widen the logits gaps between hallucinatory (*Spoon*, *Table*, and *Fork*) and targeted tokens (*People*), yielding a new hallucination as *Fork*.

To tackle this issue, we conducted the theoretical analysis to explore mechanisms for more effective contrast decoding (refer to Section 5 for detailed information on the process). Theoretically, a clear distinction between hallucinatory and target tokens can significantly enhance the effectiveness of contrast decoding methods in mitigating hallucinations. Based on this crucial insight, we introduce a novel optimization strategy called Hallucination-Induced Optimization (HIO). This strategy enhances the distinction between hallucinatory and targeted tokens by utilizing a refined theoretical preference model(as shown in the Fig. 1 on the left, section (c)), accurately outputting the correct result, *People*.

Consequently, this improves the efficiency of contrast decoding, thereby mitigating hallucinations in Large Vision-Language Models (LVLMs). Furthermore, our proposed method significantly reduces hallucinations in LVLMs compared to existing contrast decoding methods(as shown in the Fig. 1 on the right). To sum up, our main contributions are as follows:

1. We conducted a comprehensive theoretical analysis to explore mechanisms that enhance the effectiveness of the contrast decoding strategy.

2. We introduce Hallucination-Induced Optimization (HIO), an innovative strategy that utilizes a finely-tuned theoretical preference model to intensify the contrast between hallucinatory and target tokens. This enhancement strengthens the effectiveness of contrast decoding and effectively reduces hallucinations in Large Visual Language Models (LVLMs).

3. Extensive experimental research demonstrates that our Hallucination-Induced Optimization (HIO) strategy effectively reduces hallucinations in Large Visual Language Models (LVLMs), surpassing state-of-the-art methods across various benchmarks.

## 2 Related Work

**Hallucination in LVLMs.** Before the advent of Large Language Models (LLMs), "hallucination" in natural language processing (NLP) primarily referred to generating nonsensical or source-deviating content Lee et al. [2018], Zhou et al. [2020], Lin et al. [2021], Ji et al. [2023], Zhang et al. [2023], Shi et al. [2023]. Recent studies have tackled the complexities of object hallucination in Large Vision-Language Models (LVLMs), focusing on evaluation and detection methods Wang et al. [2023a], Liu et al. [2023a], Li et al. [2023b], Lovenia et al. [2023]. The CHAIR metric Rohrbach et al. [2018] evaluates the exact match between generated and ground-truth image captions, while POPE Li et al. [2023b] assesses the model's awareness of object existence through binary classification.

**Decoding Method.** The decoding method determines the generation of text tokens at each time step within language models. Traditional decoding strategies such as beam search Boulanger-Lewandowski et al. [2013], top-k decoding Fan et al. [2018], and sampling methods Holtzman et al. [2019], despite their widespread use, are prone to producing hallucinatory content. Recent research Li et al. [2022], Chuang et al. [2023], Leng et al. [2023], Huang et al. [2023] has made attempts to address this issue by proposing better decoding methods. For instance, Leng et al. [2023] uses contrastive decoding in LVLMs; However, global visual uncertainty poses challenges to the precise induction of hallucinatory tokens, limiting the effectiveness of mitigation strategies and risking unwanted hallucinations. To address this, we developed Hallucination-Induced Optimization (HIO), a novel strategy that enhances the contrast between hallucinatory and targeted tokens. Fig.1 presents the comparison results, where our approach demonstrates superior performance than other decoding methods.

## 3 Preliminaries

We first review the Contrast Decoding pipeline in Leng et al. [2023] (and later Zhang et al. [2024], Favero et al. [2024]). Then take a close look at the Bradley-Terry model Bradley and Terry [1952] and its application such as Direct Preference Optimization Rafailov et al. [2024]. Inspired by these studies, we propose our Hallucination-Induced Optimization.

**Visual Contrastive Decoding.** We consider an LVLM parameterized by $\theta$. The model takes a textual query input $x$ and a visual input $v$, where $v$ provides contextual visual information to assist the model in generating a relevant response $y$ to the textual query. The response $y$ is sampled auto-regressively from the probability distribution conditioned on the query $x$ and the visual context $v$. Mathematically, this can be formulated as:

$$y_t \sim p_\theta\left(y_t \mid v, x, y_{<t}\right) \propto \exp \operatorname{logit}_\theta\left(y_t \mid v, x, y_{<t}\right) \tag{1}$$

where $y_t$ denotes the token at time step $t$, and $y_{<t}$ represents the sequence of generated tokens up to the time step $t - 1$. Specifically, given a textual query $x$ and a visual input $v$, the model generates two distinct output distributions: one conditioned on the original $v$ and the other on the distorted visual input $v'$, which is derived by applying pre-defined distortions (i.e., Gaussian noise mask) to the original $v$. Then, a new contrastive probability distribution is computed by exploiting the differences

between the two initially obtained distributions. The new contrastive distribution $p_{vcd}$ is formulated as:

$$p_{vcd}(y \mid v, v', x) = \text{softmax}[(1 + \alpha) \text{logit}_\theta (y \mid v, x) - \alpha \text{logit}_\theta (y \mid v', x)] \quad (2)$$

where larger value of $\alpha$ indicate a stronger amplification of differences between the two distributions ($\alpha = 0$ reduces to regular decoding).

**Direct Preference Optimization.** Reinforcement learning (RL) effectively fine-tunes Large Language Models (LLMs) to align with human behavior. Given an input $x$ and a response $y$, a language model policy $\pi_\theta$ generates a conditional distribution $\pi_\theta(y \mid x)$. RL aims to maximize the average reward of outputs, with the reward function $r(x, y)$. To prevent *overoptimization* Gao et al. [2023], the objective loss includes a KL-divergence term, controlling the divergence between the language model policy and its reference policy $\pi_{\text{ref}}(y \mid x)$, typically derived from supervised fine-tuning. Thus, the overall objective is formulated as:

$$\max_{\pi_\theta} \mathbb{E}_{x \sim \mathcal{D}, y \sim \pi_\theta(y|x)} \left[ r(x, y) - \alpha \log \frac{\pi_\theta(y \mid x)}{\pi_{\text{ref}}(y \mid x)} \right] \quad (3)$$

where $\mathcal{D}$ is a dataset of prompts and $\alpha$ is a coefficient to control KL-divergence term. However, optimizing the above loss term with common strategies like proximal policy optimization (PPO) Schulman et al. [2017] is complex to tune. Recently, direct preference optimization (DPO) Rafailov et al. [2024] simplifies the above process by leveraging preference data for optimization. Here, the preference data is defined as $\mathcal{D} = \{x^{(i)}, y_w^{(i)}, y_l^{(i)}\}_{i=1}^N$, where $y_w^{(i)}$ and $y_l^{(i)}$ represent preferred and dispreferred responses given an input prompt $x$. These are then presented to human labelers who express preferences for one answer, denoted as $y_w \succ y_l \mid x$ where $y_w$ and $y_l$ denote the preferred and dispreferred respectively. Following a Bradley-Terry model [Bradley and Terry, 1952], the probability of obtaining each preference pair is:

$$p(y_w \succ y_l \mid x) = \frac{\exp(r(x, y_w))}{\exp(r(x, y_w)) + \exp(r(x, y_l))}. \quad (4)$$

where the superscript $i$ is omitted for simplicity. In DPO, the optimization of Eqn. (3) can be formulated as classification loss over the preference data as:

$$\mathcal{L}_{DPO}(\pi_\theta; \pi_{\text{ref}}) = -\mathbb{E}_{(x, y_w, y_l) \sim \mathcal{D}} \left[ \log \sigma \left( \alpha \log \frac{\pi_\theta(y_w|x)}{\pi_{\text{ref}}(y_w|x)} - \alpha \log \frac{\pi_\theta(y_l|x)}{\pi_{\text{ref}}(y_l|x)} \right) \right]. \quad (5)$$

DPO enables learning $\pi_\theta$ from a fixed dataset of preferences, which is lightweight. However, the challenge arises because the direct application of DPO does not reliably induce hallucinations in a manner that meets the criteria specified in Eqn. (17).

## 4 Method

An overview of the proposed HIO method is shown in Fig. 2. It constructs a more-hallucinated LVLM by inducing hallucinations from the original LVLM to amplify the contrast between hallucinatory and targeted tokens, thereby enhancing the efficiency of contrast decoding and mitigating hallucinations in LVLMs. In Section 4.1, we harness a fine-tuned theoretical preference model to amplify the contrast between hallucinatory and targeted tokens. Furthermore, to induce more potential hallucinations for effective contrast decoding, we propose to amplify multiple hallucination tokens based on a theoretical foundation presented in Eqn. 17 of Section 5. This theory demonstrates that effective contrastive decoding requires a consistent difference between the logits of potential hallucinated tokens and the correct token. And Section 4.3 introduces additional constraints to overcome the limitations of existing classification loss in amplifying the contrast between hallucinatory and targeted tokens.

### 4.1 Contrary Bradley-Terry Model (CBTM)

We harness a fine-tuned theoretical preference model (i.e., Contrary Bradley-Terry Model [Bradley and Terry, 1952]) to amplify the contrast between hallucinatory and targeted tokens. The studies on hallucination mitigation Zhao et al. [2023], Yu et al. [2023], Zhou et al. [2024] utilize BT model by defining the non-hallucinatory output as $y_w$ and the hallucinatory output as $y_l$. Subsequently, they employ BT model training to incentivize the model to prioritize outputs without hallucinations over

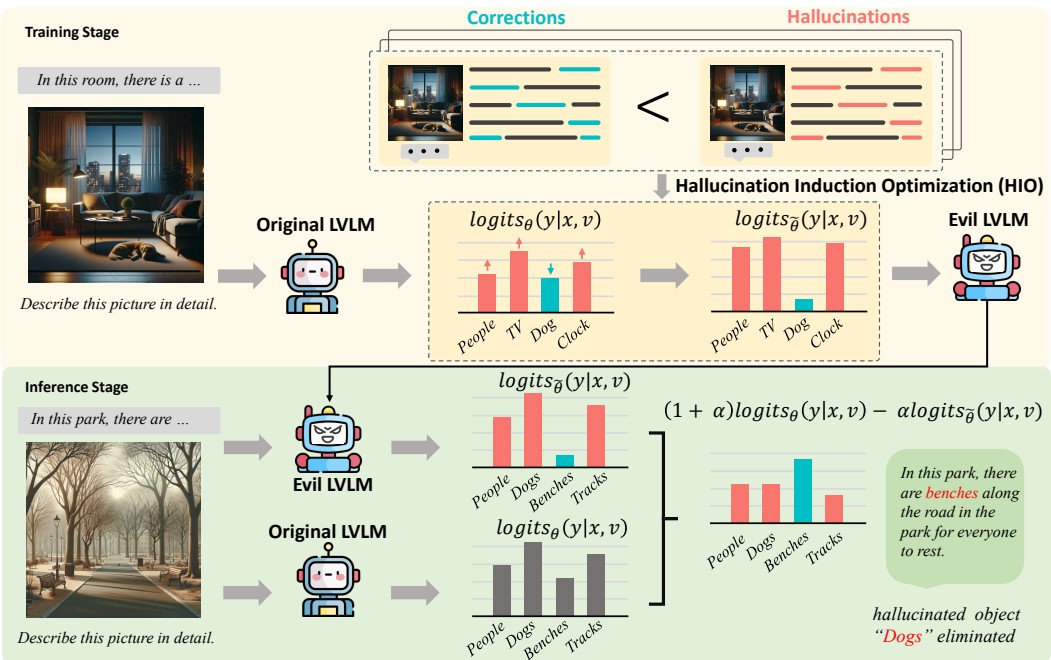

Figure 2: **An overview of Hallucination-Induced Optimization (HIO).** Our approach comprises two phases: the training stage and inference decoding. During the training stage, given an input image, a query, and a manually annotated correction, the Large Visual Language Model (LVLM) produces multiple instances of hallucinated content. We then apply our Hallucination-Induced Optimization (HIO) method to train an 'Evil' LVLM by inducing hallucinations from the original LVLM. In the inference phase, the logits from the trained 'Evil' LVLM are used to contrast with those generated by the original LVLM, effectively reducing the presence of hallucinations.

those containing them.

However, within the context of contrast decoding, inducing hallucinations is crucial, and the resulting model output must satisfy the criteria outlined in Eqn. (17). (The detailed derivation of this formula is provided in Section5). To meet the requirements specified in Eqn. (17), the logits associated with hallucinated tokens $\hat{l}_i^{\{v,x,y_{<t}\}}$ need amplification, while at least one of the logits for the correct token $\hat{l}_j^{\{v,x,y_{<t}\}}$ must be reduced. In contrast to the prevailing research efforts focused on alleviating hallucinations, our approach enables the model to learn to fit the distribution containing hallucinations while avoiding convergence with the distribution of correct outputs. The details are outlined as follows. To regulate $\hat{l}_i^{\{v,x,y_{<t}\}}$ and $\hat{l}_j^{\{v,x,y_{<t}\}}$, we utilize the dataset introduced by Yu et al. [2023]. This dataset is notable for providing a pair of outputs per input, with the output paragraphs being mostly identical except for differences in certain words or short phrases. By leveraging this dataset, we approximate the conditions outlined in Eqn. (17) within a unified statement. Different from Eqn. (5), we apply the Bradley-Terry (BT) [Bradley and Terry, 1952] model in a reversed way, the objective is:

$$
\begin{aligned}
p(y_l \succ y_w \mid x) &= \frac{\exp\left(r(x, y_l)\right)}{\exp\left(r(x, y_l)\right) + \exp\left(r(x, y_w)\right)} \\
&= \sigma\left(\beta \log \frac{\pi_\theta(y_l|v, x)}{\pi_{\text{ref}}(y_l|v, x)} - \beta \log \frac{\pi_\theta(y_w|v, x)}{\pi_{\text{ref}}(y_w|v, x)}\right).
\end{aligned}
\tag{6}
$$

where $\sigma(\cdot)$ is defined as a sigmoid function and the reference model $\pi_{\text{ref}}(y|x)$ is usually implemented by an instruction-tuned base model we want to improve, and is kept fixed during DPO training. Only the policy model $\pi_\theta(y|x)$ is updated.

## 4.2 Amplification of Multiple Targeted Hallucination (AMTH)

The methodology delineated in Eqn. (6), along with the conventional application of Direct Preference Optimization (DPO) for mitigating hallucinations, is limited to highlight the difference between a single hallucination token and the target token. Consequently, these approaches fall short in enhancing

the distinctions among other hallucinations relative to the target tokens, which is critical as shown in Eqn. (17). In this section, we will explain how to amplify the differences between multiple hallucination tokens and target tokens through modifications at both the loss function and data levels.

**Multiple Hallucination-Induced Optimization.** Achieving the desired distribution through single positive and negative sample fitting preference training is not feasible, leading conventional Direct Preference Optimization (DPO) applications Zhao et al. [2023], Yu et al. [2023], Zhou et al. [2024] to overlook a significant number of hallucinations. Thus, drawing inspiration from the implications of Eqn. (17), our approach strategically induces multiple hallucinations to increase the probability of producing a correct word in the output. As demonstrated in Eqn. (17), effective contrast decoding necessitates not only the amplification of one hallucination but also the consideration of a diverse set of potential hallucinations. We propose the simultaneous fitting of multiple pairs of preference data when modeling distributions for the same input preference, treating all pairs of preference data with equal importance. Based on Eqn. (6), we apply the Bradley-Terry (BT) [Bradley and Terry, 1952] model in a multi-pair way, the objective is:

$$
\begin{aligned}
\prod_{i=1}^{k} p(y_l \succ y_w \mid x) &= \prod_{i=1}^{k} \frac{\exp\left(r(x, y_{li})\right)}{\exp\left(r(x, y_{li})\right) + \exp\left(r(x, y_w)\right)} \\
&= \prod_{i=1}^{k} \sigma\left(\beta \log \frac{\pi_\theta(y_{li}|x)}{\pi_{\text{ref}}(y_{li}|x)} - \beta \log \frac{\pi_\theta(y_w|x)}{\pi_{\text{ref}}(y_w|x)}\right).
\end{aligned}
\tag{7}
$$

where $\{y_{li}\}, i \in \{1, 2, \ldots, k\}$ represent the multiple potential hallucination tokens. Assuming access to a static dataset of comparisons $\mathcal{D} = \left\{x^{(i)}, y_w^{(i)}, \{y_{li}^{(i)}\}\right\}_{i=1}^{N}$ sampled from $p$, we can parametrize a reward model $r(x, y)$ and estimate the parameters via maximum likelihood. Framing the problem as a binary classification we have the negative log-likelihood loss:

$$
\mathcal{L}_{\text{AMTH}}(\pi_\theta; \pi_{\text{ref}}) = -\mathbb{E}_{(x, y_l, y_w) \sim D}\left[\log\left(\prod_{i=1}^{k} p(y_l \succ y_w \mid x)\right)\right]
\tag{8}
$$

$$
= -\mathbb{E}_{(x, y_l, y_w) \sim D} \sum_{i=1}^{k}\left[\log \sigma\left(\beta \log \frac{\pi_\theta(y_{li}|v, x)}{\pi_{\text{ref}}(y_{li}|v, x)} - \beta \log \frac{\pi_\theta(y_w|v, x)}{\pi_{\text{ref}}(y_w|v, x)}\right)\right]
\tag{9}
$$

**Acquisition of Multiple Candidate Hallucinations.** While numerous hallucination datasets exist Yu et al. [2023], Zhao et al. [2023], Zhou et al. [2024], they are either generated by GPT or manually rewritten, and thus do not accurately represent the model's potential for multiple hallucinations. Therefore, we propose a novel approach: allowing the model to directly output tokens with high confidence as negative samples. While this approach may incorrectly classify some correct tokens as hallucinations, it compensates by providing true value-labeled data for correction and supplementation. Consequently, this method effectively amplifies multiple hallucinations while reducing the target token. The detailed training process of our method is outlined in Algorithm 1.

### 4.3 Advanced Constraints for Inducing (ACI)

To overcome the limitations of existing classification loss in amplifying the contrast between hallucinatory and targeted tokens, we introduces additional constraints. The preference optimization strategy outlined in Eqn. (8) allows the model to accommodate a specific range of preference distributions through the cross-entropy in the classification loss function. The precise formulation is as follows:

$$
\pi_\theta(y_l|v, x) = \sum_{t=1}^{m} \frac{\exp \hat{l}_{k_t}^{\{v, x, y_{<t}\}}}{\sum_j^N \exp \hat{l}_j^{\{v, x, y_{<t}\}}}, \{k_t\} \in y_l, t = \{1, 2, \ldots, m\}
\tag{10}
$$

where $m$ represents the length of the sentence $y_l$ and $\{k_T\}$ is token of each word, and the definition of $\hat{l}_i^{\{v, x, y_{<t}\}}$ is shown in Section 5. While the use of cross-entropy to minimize encoding length helps the model align with the desired output sentence, it does not consistently ensure that the logits of induced hallucinations meet the conditions specified in Eqn. (17).

For example, the goal of Eqn. (8) is to increase $\pi_\theta(y_l|v, x)$, but both increasing $\exp \hat{l}_{k_t}^{\{v,x,y_{<t}\}}$ or decreasing $\sum_j^N \exp \hat{l}_j^{\{v,x,y_{<t}\}}$ can achieve this goal. Meanwhile, decreasing the value of $\sum_j^N \exp \hat{l}_j^{\{v,x,y_{<t}\}}$ can also allow $\pi_\theta(y_w|v, x)$ to meet the optimization criteria. As shown in Fig. 3, the blue curve, representing the disparity between the logits of the hallucinatory and targeted tokens, typically exhibits a positive trend. Nevertheless, it's important to note occasional segments where this value dips below zero. To tackle this issue, we further add restrictions based on Eqn. (8):

$$\mathcal{L}_{\text{HIO}}(\pi_\theta; \pi_{\text{ref}}) = -\mathbb{E}_{(x,y_l,y_w)\sim D} \sum_{i=1}^{k} \left[ \log \sigma \left( \beta \log \frac{\pi_\theta(y_{li}|v, x)}{\pi_{\text{ref}}(y_{li}|v, x)} - \beta \log \frac{\pi_\theta(y_w|v, x)}{\pi_{\text{ref}}(y_w|v, x)} \right) \right. \tag{11}$$
$$\left. + \gamma \left( \frac{1}{m} \sum_{t=1}^{m} \hat{l}_{k_t}^{\{v,x,y_{<t}\}} - \hat{l}_i^{\{v,x,y_{<t}\}} \right) \right]$$

By implementing this constraint, the model can be fitted to the distribution of preference statements, thereby further expanding the difference between hallucination tokens and target tokens.

## 5 Fundamental Conditions for Contrast Decoding

Contrast decoding is capable of mitigating hallucinations when specific conditions are met. This section delves into a comprehensive discussion and analysis of these conditions.

**Definition.** Let $l_i^{\{v,x,y_{<t}\}}$ represent the probability of the $i$-th token in the model's vocabulary given the query $x$, the visual context $v$ and the sequence of generated tokens up to the time step $(t-1)$. The logits can be formulated as:

$$\text{logit}_\theta(y_t \mid v, x, y_{<t}) = L^{\{v,x,y_{<t}\}} = (l_1^{\{v,x,y_{<t}\}}, l_2^{\{v,x,y_{<t}\}}, \ldots, l_N^{\{v,x,y_{<t}\}}) \tag{12}$$

where $N$ denotes the vocabulary length.

**Definition.** Let $\hat{L}^{\{v,x,y_{<t}\}}$ represents the ideal logits for contrast decoding, $L'^{\{v,x,y_{<t}\}}$ represents the logits with hallucination and $L^{*\{v,x,y_{<t}\}}$ represents the logits of correct token, where $\{L'^{\{v,x,y_{<t}\}}, L^{*\{v,x,y_{<t}\}}\} \in L^{\{v,x,y_{<t}\}}$. The results of contrast decoding of logits can be formulated as:

$$\delta^{\{v,x,y_{<t}\}} = (1+\alpha)L^{\{v,x,y_{<t}\}} - \alpha\hat{L}^{\{v,x,y_{<t}\}} \tag{13}$$

where larger $\alpha$ values indicate a stronger amplification of differences between the two distributions ($\alpha = 0$ reduces to regular decoding). The condition for the absence of hallucination in the logits subsequent to subtraction is that the values of the logits corresponding to all hallucinatory tokens are less than the magnitudes of the logits corresponding to the correct lexical tokens. The aforementioned condition is articulated mathematically as follows:

**Proposit.**

$$\max \delta'^{\{v,x,y_{<t}\}} < \min \delta^{*\{v,x,y_{<t}\}} \tag{14}$$

where $\delta'^{\{v,x,y_{<t}\}}$ denotes the result of the subtraction between the logits of all hallucinated vocabulary tokens and the logits after their ideal amplification. $\delta^{*\{v,x,y_{<t}\}}$ represents the outcome of the subtraction between the logits corresponding to all correct vocabulary tokens and the logits under the ideal scenario. Eqn. 14 represents a theoretical upper bound, which guides us in enhancing the effectiveness of Contrast Decoding method for hallucination elimination by ensuring that the logits of all hallucinated words are lower than those of the correct words. Upon expansion of the left side of the equation, the following result is obtained:

$$\max \delta'^{\{v,x,y_{<t}\}} = \max\{(1+\alpha)L'^{\{v,x,y_{<t}\}} - \alpha\hat{L}'^{\{v,x,y_{<t}\}}\}$$
$$= \max\{(1+\alpha)l_i^{\{v,x,y_{<t}\}} - \alpha\hat{l}_i^{\{v,x,y_{<t}\}}\}, i \in \{k_1', k_2', \ldots, k_m'\} \tag{15}$$
$$\geq \frac{1}{m} \sum_{i=k_1}^{k_m} ((1+\alpha)l_i^{\{v,x,y_{<t}\}} - \alpha\hat{l}_i^{\{v,x,y_{<t}\}})$$

where $m$ denotes the total number of hallucinated vocabulary items, and $k_j$ represents the subscript position of the $i$-th hallucinated vocabulary within the set $L^{\{v,x,y_{<t}\}}$. For the right side of the equation,

one of the correct lexical items is selected as the subject for amplification.

$$\min \delta^{*\{v,x,y_{<t}\}} = \min\{(1+\alpha)L^{*\{v,x,y_{<t}\}} - \alpha\hat{L}^{*\{v,x,y_{<t}\}}\}$$
$$\leq (1+\alpha)l_j^{\{v,x,y_{<t}\}} - \alpha\hat{l_j}^{\{v,x,y_{<t}\}}, j \in \{k_1^*, k_2^*, \ldots, k_n^*\} \tag{16}$$

where $n$ denotes the total number of correct lexical items. Based on Eqn. (15) and Eqn. (16), Eqn. (14) can be simplified to the form presented as follows:

$$m \times ((1+\alpha)l_j^{\{v,x,y_{<t}\}} - \alpha\hat{l_j}^{\{v,x,y_{<t}\}}) - \sum_{i=k_1}^{k_m}((1+\alpha)l_i^{\{v,x,y_{<t}\}} - \alpha\hat{l_i}^{\{v,x,y_{<t}\}}) > 0$$
$$\sum_{i=k_1}^{k_m}(\hat{l_i}^{\{v,x,y_{<t}\}} - \hat{l_j}^{\{v,x,y_{<t}\}}) > J \tag{17}$$

where $J$ represents $\frac{(1+\alpha)}{\alpha}\sum_{i=k_1}^{k_m}(l_i^{\{v,x,y_{<t}\}} - l_j^{\{v,x,y_{<t}\}})$. In the context of the contrast decoding method, given that the parameters of the original model remain invariant, the output can be characterized as a constant. Eqn. 17 delineates the logits for all hallucinated tokens $\hat{l_i}^{\{v,x,y_{<t}\}}$ and contrasts these with the logits of a single correct token $\hat{l_j}^{\{v,x,y_{<t}\}}$. It postulates that, for an optimal logits output, a pronounced divergence must be maintained between the logits of hallucinated tokens and the logit of the correct token.

Eqn. 17 illustrates that hallucinations can be effectively eliminated through contrastive decoding if the difference between the logits of the hallucinatory token and the correct token in the 'Evil' LVLM's output (Left part of Eqn.17) exceeds that in the original LVLM output ($J$ in Eqn.17). For example, as depicted in the lower part of Fig. 2, where "Dogs" is a hallucination and "Benches" is the correct label, the hallucination of "Dogs" is removed when the difference between the logits for "Dogs" and "Benches" in the 'Evil' LVLM output surpasses the difference in the original LVLM output. When this condition is met for all potential hallucinations, all hallucinations are effectively eliminated.

# 6 Experiments

## 6.1 Experimental Settings

**Benchmarks.** We evaluate HIO on three benchmarks including: (1) Quantitative metrics POPE Li et al. [2023b] on MSCOCO Lin et al. [2014] dataset. The Polling-based Object Probing Evaluation Li et al. [2023b] offers a streamlined approach to assessing object hallucination. In this benchmark, LVLMs are queried about the existence of specific objects in a given image. (2) CHAIR Rohrbach et al. [2018], Caption Hallucination Assessment with Image Relevance, is a specialized tool designed to evaluate the occurrence of object hallucination in image captioning tasks. (3) General-purpose Multimodal Large Language Model Evaluation (MME) Fu et al. [2023] benchmark, which provides an extensive benchmark designed to evaluate LVLMs across multiple dimensions, including ten perception-related subtasks and four cognition-focused ones.

**Implementation Description** We evaluate our model across three Large Vision-Language Models (LVLMs): LLaVA 1.5, InstructBLIP, and MiniGPT-4. For decoding, we use Llama-7B and Vicuna-7B as the linguistic decoder for LLaVA and InstructBLIP/MiniGPT-4, respectively. Our model's performance is compared against three leading models in the field: OPERA Huang et al. [2023], VCD Leng et al. [2023], and VDD Zhang et al. [2024]. To ensure a fair and rigorous comparison, we adhere to the configurations and guidelines from the original works and codebases of the compared models. The training is conducted on a robust computational setup: 4x RTX 3090 GPUs for LLaVA 1.5, 8x V100 GPUs for MiniGPT-4, and 4x A6000 GPUs for InstructBLIP. Each training session lasts approximately 2-4 hours. Hyperparameters including alpha and beta are set to 1.0 and 0.1, respectively, in accordance with the VCD model's specifications.

## 6.2 Experimental Results

**POPE.** To evaluate HIO's capability on object hallucination, we compare it with several state-of-the-art Decoding methods on POPE. The results are shown in Tab. 1, which presents the experimental

results on the POPE dataset across random, popular, and adversarial settings. Our method consistently outperforms the standard decoding strategy, with average improvements of 6.2% in accuracy and 7.3% in F1 score across all LVLMs. Additionally, our approach clearly surpasses state-of-the-art decoding methods, demonstrating its effectiveness in mitigating object hallucinations. The improved performance across *random*, *popular*, and *adversarial* settings further confirms that our HIO method effectively reduces hallucinations in diverse scenario.

| Dataset | Setting | Decoding | Accuracy↑ | Precision | Recall | F1 Score↑ |
|---------|---------|----------|-----------|-----------|--------|-----------|
| MSCOCO | *Random* | Regular | 83.29 | 92.13 | 72.80 | 81.33 |
| | | VCD | 87.73 | 91.42 | 72.80 | 87.16 |
| | | ICD | 89.56 | 88.71 | 90.66 | 89.68 |
| | | VDD | 90.00 | 97.36 | 79.13 | 88.79 |
| | | Ours | **90.21** | **93.23** | **86.85** | **89.94** |
| | *Popular* | Regular | 81.88 | 88.93 | 72.80 | 80.06 |
| | | VCD | 85.38 | 86.92 | 83.28 | 85.06 |
| | | ICD | 86.16 | 83.18 | 90.66 | 86.76 |
| | | VDD | 85.91 | 94.33 | 76.33 | 84.40 |
| | | Ours | **88.12** | 88.96 | **86.83** | **87.84** |
| | *Adversarial* | Regular | 78.96 | 83.06 | 72.75 | 77.57 |
| | | VCD | 80.88 | 79.45 | 83.29 | 81.33 |
| | | ICD | 79.71 | 74.35 | 90.66 | 81.70 |
| | | VDD | 83.52 | 89.34 | 76.20 | 82.20 |
| | | Ours | **84.32** | 84.28 | **84.33** | **84.34** |

Table 1: Results on POPE. *Regular* decoding denotes direct sampling, whereas *VCD* refers to Visual Contrastive Decoding method, whereas *VDD* refers to Visual Debias Decoding. The best performances within each setting are **bolded**.

**CHAIR.** Beyond the "Yes-or-No" discriminative evaluations conducted on the POPE and MME datasets, we also assess our model's performance in open-ended caption generation using the CHAIR benchmark. Tab.2 and Tab.5 display results for 500 randomly selected images from the COCO val2017 and val2014 datasets, respectively. These results show consistent improvements in our model compared to other methods. Specifically, our approach significantly reduces object hallucinations in generated captions, as evidenced by lower CHAIRS and CHAIRI scores (8.1% reduction in CHAIRS and 4.9% in CHAIRI). Furthermore, it enhances caption detail, as indicated by higher Recall scores. Overall, our method achieves an effective balance between accuracy and detail in open-ended caption generation by widening the gap between hallucinated and correct tokens.

| Row | Method | Length | $\text{CHAIR}_S \downarrow$ | $\text{CHAIR}_I \downarrow$ | Recall ↑ |
|-----|--------|--------|----------|----------|--------|
| 1 | - | 100.6 | 50.0 | 15.4 | 77.1 |
| 2 | VCD | 100.4 | 48.6 | 14.9 | 77.3 |
| 3 | OPERA | 98.6 | 47.8 | 14.6 | 76.8 |
| 4 | OPERA (fast) | 85.3 | 48.6 | 14.5 | 76.7 |
| 5 | ICD | 106.3 | 50.8 | 15.0 | 78.5 |
| 6 | **Ours** | 110.3 | **41.4** | **10.5** | **77.4** |

Table 2: Hallucination performance of different methods.

**MME.** To evaluate HIO's capability on object-level and attribute-level hallucination, we compare it with several state-of-the-art Decoding methods on MME. The results are shown in Tab. 3. Consistent with the performance on POPE and CHAIR, HIO also achieves competitive results on MME compared to other decoding methods. Concretely, HIO outperforms the VCD 6.4%, 21.7%, 4.7% and 17.0% at *Existence*, *Count*, *Position* on MME, respectively. The results demonstrate the effectiveness of our method.

## 6.3 Ablation Study

To verify the effectiveness of each component of the proposed HIO, we conduct ablation studies on Contrary Bradley-Terry Model(CBTM), Amplification of Multiple Targeted Hallucination(AMTH) and Advanced Constraints for Inducing(ACI) under the MSCOCO Lin et al. [2014]. The results are shown in Tab. 4. when constrained by CBTM in Exp 2, the model outperforms the baseline(*i.e.,* Exp 1). This helps LVLM amplify hallucinations. Furthermore, after being integrate with AMTH

| Model | Decoding | Object-level | | Attribute-level | | Total Scores↑ |
|---|---|---|---|---|---|---|
| | | *Existence↑* | *Count↑* | *Position↑* | *Color↑* | |
| LLaVA1.5 | Regular | 175.67 | 124.67 | 114.00 | 151.00 | 565.33 |
| | VCD | 184.66 | 138.33 | 128.67 | 153.00 | 604.66 |
| | VDD | 190.00 | 143.33 | 145.00 | 165.00 | 643.33 |
| | Ours | **190.00** | **160.00** | 133.33 | **170.00** | **653.33** |

Table 3: Results on the hallucination subset of MME. Regular decoding denotes direct sampling, *VCD* denotes Visual Contrastive Decoding method, whereas *VDD* refers to Visual Debias Decoding. The best performances within each setting are **bolded**.

in Exp 3, LVLM obtain sigificant gains on $\text{CHAIR}_S$ and $\text{CHAIR}_I$. When integrate with ACI, the LVLM achieve superior performance on $\text{CHAIR}_S$, $\text{CHAIR}_I$ and Recall. These results demonstrate the effective of each component. Moreover, we have enriched the ablation study to analyze the

| Exp | CBTM | AMTH | ACI | $\text{CHAIR}_S \downarrow$ | $\text{CHAIR}_I \downarrow$ | Recall↑ |
|---|---|---|---|---|---|---|
| 1 | - | - | - | 33.4 | 9.07 | 81.1 |
| 2 | ✓ | - | - | 18.6 | 5.08 | 79.9 |
| 3 | ✓ | ✓ | - | 14.2 | 3.06 | 80.5 |
| 4 | ✓ | ✓ | ✓ | **11.2** | **2.02** | **81.3** |

Table 4: Ablation study with different components of our model on CHAIR-COCO.

generalization capability of our proposed components to unseen categories, as detailed in Table 4. For the Unseen-P dataset, we collected data from MSCOCO, A-OKVQA, and GQA, ensuring no overlap with the training set, resulting in 495 samples across 10 distinct classes. These experiments show that our components generalize effectively to unseen data. Finally, we have integrated the ablation study into the experimental results section, rather than presenting it separately.

| Dataset | CBTM | AMTH | ACI | Accuray ↑ | $\text{Precision}_I$ ↑ | Recall ↑ | F1 Score ↑ |
|---|---|---|---|---|---|---|---|
| unseen-N | - | - | - | 88.88 | 84.88 | 95.63 | 83.93 |
| | ✓ | - | - | 89.79 | 86.22 | **95.63** | 90.68 |
| | ✓ | ✓ | - | 91.83 | **95.30** | 88.64 | 91.85 |
| | ✓ | ✓ | ✓ | **92.97** | 91.94 | 94.75 | **93.33** |
| unseen-P | - | - | - | 81.15 | 64.86 | 100.00 | 78.68 |
| | ✓ | - | - | 82.61 | 66.66 | **100.00** | 80.02 |
| | ✓ | ✓ | - | 84.05 | 72.41 | 87.51 | 79.24 |
| | ✓ | ✓ | ✓ | **85.51** | **75.01** | 87.51 | **80.76** |

Table 5: Ablation study on the generalization of each component on unseen datasets.

# 7 Discussion

In this study, we conduct an in-depth examination of the principles governing contrast decoding and the prerequisites for its efficacy. Based on our findings, we introduce HIO, an innovative model optimization approach designed to induce hallucinations. This method significantly amplifies hallucinatory elements within the model, thereby effectively mitigating them through contrast decoding. Extensive experimentation across various datasets has demonstrated that HIO effectively reduces hallucinations and achieves state-of-the-art performance.

**Limitations & Future Work.**
Our findings establish a necessary, but not sufficient, condition for the successful operation of contrast decoding. Further exploration of more effective conditions could significantly enhance the efficiency of contrast decoding in mitigating hallucinations. Additionally, exploring training-free methods to induce hallucinations could reduce the computational costs associated with decoding.

# 8 Acknowledgments and Disclosure of Funding

This study is supported by grants from the National Natural Science Foundation of China (Grant No. 62122018, No. 62020106008, No. U22A2097, No. U23A20315), and Kuaishou, and Natural Science Foundation of Sichuan Province (Grant No. 2025ZNSFSC1463).

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

# Appendix

## A  Algorithm

The algorithm outlines the process by which the model generates its own series of potential hallucinations. Using the sample pairs produced by the model, we apply our proposed Hallucination-Induced Optimization (HIO) to enhance the distinction between hallucinated and target labels. Ultimately, hallucinations are mitigated through contrastive decoding.

Using paired hallucination and non-hallucination annotations from the RLHF-V dataset, we apply beam search to generate multiple outputs where hallucination token annotations occur. These outputs include both correct and hallucinated results, which we use as hallucination samples to reinforce the model's confidence in its outputs. The correct annotations from RLHF-V serve as ground truth, helping the model avoid hallucinations by differentiating between hallucinated and target tokens. This approach expands the contrast between hallucinated and target tokens, effectively reducing hallucinations.

---

**Algorithm 1** Training LVLM to Amplify Multiple Targeted Hallucination

---

**Require:** training image set $\mathcal{V}$; user prompt set $\mathcal{X}$; pair-wise groundtruth descriptions, $\mathcal{Y}'$ for hallucination description and $\mathcal{Y}^*$ for correct description; LVLM $\mathcal{M}(\cdot)$ with parameters $\theta$

1: According to each pair's hallucination description $\mathcal{Y}'$ and correct description $\mathcal{Y}^*$, get starting subscripts of $\mathcal{Y}'$ compared with $\mathcal{Y}^*$. Different subscripts denoted as $\mathcal{I} = \{i'_1, i'_2, \ldots, i'_n\}$.

2: Initialize the LVLM's parameter $\theta$ and an empty set $\mathcal{S}_{new} \leftarrow \{\}$

3: **for** each image $v \in \mathcal{V}$, each prmopt $x \in \mathcal{X}$, the correpsonding hallucinatory description $y' \in \mathcal{Y}'$ and correpsonding hallucinatory description $y^* \in \mathcal{Y}^*$ **do**

4:    Get starting subscripts of $\mathcal{Y}'$ compared with $\mathcal{Y}^*$. Different subscripts denoted as $\mathcal{I}' = \{i'_1, i'_2, \ldots, i'_m\}$

5:    **for** $i'_t \in \mathcal{I}'$ **do**

6:       $y'_{<i'_t}$ represents the sequence of generated tokens up to the time step $(i'_t - 1)$

7:       Generate next logits $L^{\{v,x,y_{<i'_t}\}} = \mathcal{M}(v, x, y'_{<i'_t}) = (l_1^{\{v,x,y_{<i'_t}\}}, l_2^{\{v,x,y_{<i'_t}\}}, \ldots, l_N^{\{v,x,y_{<i'_t}\}})$

8:       Find Top-K subscripts $J^{\{v,x,y_{<i'_t}\}} = \arg\min_{T \subseteq \{1,2,\ldots,n\}, |T|=K} \sum_{j \in T} l_j^{\{v,x,y_{<i'_t}\}} = \{j_1, j_2, \ldots, j_k\}$ where $l_{j_1}^{\{v,x,y_{<i'_t}\}} \geq l_{j_2}^{\{v,x,y_{<i'_t}\}} \geq \cdots \geq l_{j_k}^{\{v,x,y_{<i'_t}\}}$

9:       **for** $j_t \in J^{\{v,x,y_{<i'_t}\}}$ **do**

10:          $y'_{<(i'_t+1)} = y'_{<i'_t} \cup j_t$

11:          $\delta = 1$

12:          **while** $y'_{(i'_t+\delta)}$ is not period **do**

13:             $L^{\{v,x,y_{<i'_t+\delta}\}} = \mathcal{M}(v, x, y'_{<i'_t+\delta})$

14:             $y'_{<(i'_t+\delta+1)} = y'_{<i'_t+\delta} \cup \arg\min_j L^{\{v,x,y_{<i'_t+\delta+1}\}}$

15:             $\delta = \delta + 1$

16:          **end while**

17:       **end for**

18:    **end for**

19: **end for**

---

## B  Mathematical Derivations

In this appendix, we present a comprehensive verification of Eqn. (17), which is elucidated through the following detailed procedure:

$$m \times ((1+\alpha)l_j^{\{v,x,y_{<t}\}} - \alpha\hat{l}_j^{\{v,x,y_{<t}\}}) - \sum_{i=k_1}^{k_m}((1+\alpha)l_i^{\{v,x,y_{<t}\}} - \alpha\hat{l}_i^{\{v,x,y_{<t}\}}) > 0$$

$$\alpha \sum_{i=k_1}^{k_m}(\hat{l}_i^{\{v,x,y_{<t}\}} - \hat{l}_j^{\{v,x,y_{<t}\}}) - (1+\alpha)\sum_{i=k_1}^{k_m}(l_i^{\{v,x,y_{<t}\}} - l_j^{\{v,x,y_{<t}\}}) > 0$$

$$\frac{\alpha}{(1+\alpha)}\sum_{i=k_1}^{k_m}(\hat{l}_i^{\{v,x,y_{<t}\}} - \hat{l}_j^{\{v,x,y_{<t}\}}) > \sum_{i=k_1}^{k_m}(l_i^{\{v,x,y_{<t}\}} - l_j^{\{v,x,y_{<t}\}})$$

$$\sum_{i=k_1}^{k_m}(\hat{l}_i^{\{v,x,y_{<t}\}} - \hat{l}_j^{\{v,x,y_{<t}\}}) > J$$

(18)

## C  Visualization

This figure demonstrates the effectiveness of our ACI method (described in Section 4.3). The y-axis shows the difference between the hallucination token and the target token. The blue curve represents this difference without ACI, while the orange curve represents it with our proposed ACI. Clearly, our method accurately induces hallucinations, further amplifies the difference between the hallucination token and the target token, and thus effectively reduces hallucinations.

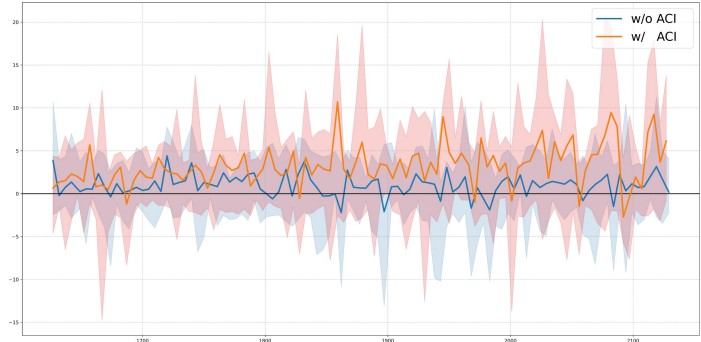

Figure 3: **The Difference between hallucination token and target token.** The horizontal axis represents the progression of training steps, while the vertical axis quantifies the disparity in logits, calculated as the hallucination token's logits minus those of the target token. It is evident that ACI effectively augments the distinction between the hallucination and target tokens.

## D  Additional experiments

**MME.** To evaluate HIO's capability on object-level and attribute-level hallucination, we compare it with several state-of-the-art Decoding methods on MME. The results are shown in Tab. 3. Consistent with the performance on POPE and CHAIR, HIO also achieves competitive results on MME compared to other decoding methods. Concretely, HIO outperforms the VCD at *Existence*, *Count*, *Position*, *Color*, *Posters*, on MME, respectively. The complete POPE evaluation is shown in the Tab 7.

| Model | Decoding | *Existence* | *Count* | *Position* | *Color* | *Posters* | *Celebrity* | Scene | Landmark | Artwork | OCR | *Percetion* |
|---|---|---|---|---|---|---|---|---|---|---|---|---|
| | Regular | 175.67 | 124.67 | 114.00 | 151.00 | 127.82 | 113.59 | 148.30 | 129.95 | 102.20 | 92.00 | 1279.19 |
| LLaVA1.5 | VCD | 184.66 | 138.33 | 128.67 | 153.00 | 132.11 | 120.94 | 152.20 | 140.45 | 109.60 | 104.00 | 1363.96 |
| | Ours | **190.00** | **160.00** | **133.33** | **170.00** | **145.50** | **138.50** | **158.70** | **165.00** | **121.00** | **142.50** | **1524.70** |

Table 6: Results on all MME perception-related tasks. The best performance of each setting is **bolded**.

| Dataset | Setting | Model | Decoding | Accuracy↑ | Precision | Recall | F1 Score↑ |
|---|---|---|---|---|---|---|---|
| MSCOCO | Random | LLaVA1.5 | Regular | 83.29 | 92.13 | 72.80 | 81.33 |
| | | | VCD | 87.73 | 91.42 | 83.28 | 87.16 |
| | | | Ours | **90.21** | **93.23** | **86.85** | **89.94** |
| | | miniGPT4 | Regular | 67.04 | 69.06 | 66.54 | 67.77 |
| | | | VCD | 69.60 | 72.76 | 66.73 | 69.62 |
| | | | Ours | **77.96** | **74.15** | **85.86** | **79.57** |
| | | InstructBLIP | Regular | 80.71 | 81.67 | 79.19 | 80.41 |
| | | | VCD | 84.53 | 88.55 | 79.32 | 83.68 |
| | | | Ours | **87.33** | **96.12** | 77.73 | **85.95** |
| | Popular | LLaVA1.5 | Regular | 81.88 | 88.93 | 72.80 | 80.06 |
| | | | VCD | 85.38 | 86.92 | 83.28 | 85.06 |
| | | | Ours | **88.1** | **88.96** | **86.83** | **87.84** |
| | | miniGPT4 | Regular | 60.89 | 61.34 | 65.74 | 63.46 |
| | | | VCD | 62.91 | 63.69 | 64.81 | 64.24 |
| | | | Ours | **72.51** | **67.75** | **85.86** | **75.74** |
| | | InstructBLIP | Regular | 78.22 | 77.87 | 78.85 | 78.36 |
| | | | VCD | 81.47 | 82.89 | 79.32 | 81.07 |
| | | | Ours | **84.83** | **90.59** | 77.72 | **83.67** |
| | Adversarial | LLaVA1.5 | Regular | 78.96 | 83.06 | 72.75 | 77.57 |
| | | | VCD | 80.88 | 79.45 | 83.29 | 81.33 |
| | | | Ours | **84.32** | **84.33** | **84.34** | **84.34** |
| | | miniGPT4 | Regular | 59.42 | 59.64 | 64.45 | 61.95 |
| | | | VCD | 62.07 | 62.15 | 66.76 | 64.37 |
| | | | Ours | **67.52** | **62.79** | **85.86** | **72.64** |
| | | InstructBLIP | Regular | 75.84 | 74.30 | 79.03 | 76.59 |
| | | | VCD | 79.56 | 79.67 | 79.39 | 79.52 |
| | | | Ours | **82.96** | **86.82** | 77.70 | **82.02** |
| A-OKVQA | Random | LLaVA1.5 | Regular | 83.45 | 87.24 | 78.36 | 82.56 |
| | | | VCD | 86.15 | 85.18 | 87.53 | 86.34 |
| | | | Ours | **90.61** | **94.97** | 85.73 | **90.19** |
| | | miniGPT4 | Regular | 64.79 | 65.26 | 65.73 | 65.50 |
| | | | VCD | 66.68 | 66.47 | 68.21 | 67.33 |
| | | | Ours | **74.74** | **69.46** | **88.13** | **77.69** |
| | | InstructBLIP | Regular | 80.91 | 77.97 | 86.16 | 81.86 |
| | | | VCD | 84.11 | 82.21 | 87.05 | 84.56 |
| | | | Ours | **88.56** | **90.25** | 86.46 | **88.32** |
| | Popular | LLaVA1.5 | Regular | 79.90 | 80.85 | 78.36 | 79.59 |
| | | | VCD | 81.85 | 78.60 | 87.53 | 82.82 |
| | | | Ours | **86.93** | **87.84** | 85.73 | **86.77** |
| | | miniGPT4 | Regular | 60.75 | 60.67 | 68.84 | 64.50 |
| | | | VCD | 62.22 | 62.23 | 68.55 | 65.24 |
| | | | Ours | **62.83** | 58.54 | **88.13** | **70.35** |
| | | InstructBLIP | Regular | 76.19 | 72.16 | 85.28 | 78.17 |
| | | | VCD | 79.78 | 76.00 | 87.05 | 81.15 |
| | | | Ours | **81.16** | **78.17** | 86.46 | **82.11** |
| | Adversarial | LLaVA1.5 | Regular | 74.04 | 72.08 | 78.49 | 75.15 |
| | | | VCD | 74.97 | 70.01 | 87.36 | 77.73 |
| | | | Ours | **80.83** | **78.08** | 85.73 | **82.71** |
| | | miniGPT4 | Regular | 58.88 | 58.56 | 68.50 | 63.14 |
| | | | VCD | 60.67 | 60.56 | 68.47 | 64.28 |
| | | | Ours | 58.36 | 55.24 | **88.24** | **67.93** |
| | | InstructBLIP | Regular | 70.71 | 65.91 | 85.83 | 75.56 |
| | | | VCD | 74.33 | 69.46 | 86.87 | 77.19 |
| | | | Ours | **74.55** | **69.74** | 86.46 | **77.22** |
| GQA | Random | LLaVA1.5 | Regular | 83.73 | 87.16 | 79.12 | 82.95 |
| | | | VCD | 86.65 | 84.85 | 89.24 | 86.99 |
| | | | Ours | **89.06** | **93.53** | 83.93 | **88.47** |
| | | miniGPT4 | Regular | 65.13 | 65.38 | 66.77 | 66.07 |
| | | | VCD | 67.08 | 68.30 | 69.04 | 68.67 |
| | | | Ours | **73.83** | **70.03** | **83.21** | **76.05** |
| | | InstructBLIP | Regular | 79.65 | 77.14 | 84.29 | 80.56 |
| | | | VCD | 83.69 | 81.84 | 86.61 | 84.16 |
| | | | Ours | **87.26** | **89.09** | 84.93 | **86.96** |
| | Popular | LLaVA1.5 | Regular | 78.17 | 77.64 | 79.12 | 78.37 |
| | | | VCD | 80.73 | 76.26 | 89.24 | 82.24 |
| | | | Ours | **84.76** | **85.35** | 83.93 | **84.63** |
| | | miniGPT4 | Regular | 57.19 | 58.55 | 60.81 | 59.66 |
| | | | VCD | 62.14 | 61.14 | 72.26 | 66.24 |
| | | | Ours | **64.74** | 60.72 | **83.28** | **70.21** |
| | | InstructBLIP | Regular | 73.87 | 69.63 | 84.69 | 76.42 |
| | | | VCD | 78.57 | 74.62 | 86.61 | 80.17 |
| | | | Ours | 77.11 | 73.42 | 84.93 | 78.76 |
| | Adversarial | LLaVA1.5 | Regular | 75.08 | 73.19 | 79.16 | 76.06 |
| | | | VCD | 76.09 | 70.83 | 88.75 | 78.78 |
| | | | Ours | **82.11** | **80.96** | 83.93 | **82.42** |
| | | miniGPT4 | Regular | 56.75 | 56.26 | 67.99 | 61.57 |
| | | | VCD | 57.78 | 57.70 | 69.82 | 63.18 |
| | | | Ours | **59.09** | 56.11 | **83.23** | **67.02** |
| | | InstructBLIP | Regular | 70.56 | 66.12 | 84.33 | 74.12 |
| | | | VCD | 75.08 | 70.59 | 85.99 | 77.53 |
| | | | Ours | 74.86 | 70.69 | 84.93 | 77.16 |

Table 7: Results on POPE. *Regular* decoding denotes direct sampling. Higher accuracy and F1 score indicate better performance and fewer hallucinations. The best performances within each setting are **bolded**.

