# OpenReview forum: "Alleviating Hallucinations in Large Vision-Language Models through Hallucination-Induced Optimization"
_NeurIPS.cc/2024/Conference — NeurIPS 2024 poster_

### Official Review · Reviewer_6ngE · 2024-07-08

**Soundness:** 3
**Presentation:** 3
**Contribution:** 2
**Rating:** 6
**Confidence:** 4

**Summary:**

This paper addresses object hallucination in large visual language models (LVLMs), a common phenomenon that such models generate texts not consistent with images. A viable approach for this issue is contrastive decoding. By comparing logits derived from images and distorted images, visual contrastive decoding reduces statistical bias and uni-modal priors thus alleviates hallucination. This paper points out that distorting images induces uncertainty which leads to unexpected results. To overcome this issue, this paper proposes to train an evil LVLM first by direct preference optimization, as illustrated in Figure 2, and contrast outputs from this model with LVLM responses. Theoretical foundations have been given and experimental results show that such strategy alleviates object hallucinations on multiple benchmarks, including POPE and CHAIR, which are commonly used metrics in this field.

**Strengths:**

- The method is simple and easy to follow.
- Contrastive decoding is to contrast outputs from a model with an amateur model. It is an interesting perspective to train an amateur model first using DPO approach.
- Writing and presentations are clear.
- The performance gains against previous methods look good.

**Weaknesses:**

- **Generalization issue**. In Figure 2, if I understand correctly, this paper induces an LVLM to generate hallucinated outputs (which is referred to as Hallucination Induction Optimization). Both hallucinated and corrected responses are involved in this Figure. If thinking more over objects (e.g., people, tv, dog, clock in Figure 2), a potential issue for this is, trained model in such manner can only be applied to these objects. In proposed hallucination-induced optimization, an LVLM is taught to hallucinate on some objects, these objects can work for later contrastive decoding. But I doubt that such capability can generalize to other objects which are not seen during training, making this a close-vocabulary approach, just like post-hoc correction methods in this field.
- Given that proposed hallucination-induced optimization resembles RLHF methods [A] in technical pipeline. Authors are suggested to include some results comparing these methods. Whether an amateur model can help an RLHF model seems an interesting question and may improve this paper.

[A] Aligning Large Multimodal Models with Factually Augmented RLHF.

**Questions:**

- For Table 1 where POPE is evaluated, I notice that previous approach VCD has three evaluation setups, including MSCOCO, GQA and A-OKVQA. Only MSCOCO is given. Authors are suggested to include the other two as well for more comprehensive evaluations.
- Typos. In line 135, An overview of the proposed HIO method is shown in Fig. 3 (3 should be 2).

**Limitations:**

Authors have indicated limitations in Appendix D Discussion.

---

> ### Author Rebuttal · Authors · 2024-08-07
>
> # Response to Reviewer 6ngE
> We thank all the reviewers for the insightful comments and the recognition of our work
>
> **"Novel Idea"** (Reviewers #1, #3, #4)
> **"Promising Results"**(Reviewers #2, $3, #4)
> **"Interesting"**(Reviewers #1, #4)
> **"Easy to understand"**(Reviewers #1, #4). Now, we carefully answer your question as follows.
>
> #### W1: Generalization issue.
> **W1-A1:**
> Our proposed Hallucination-Induced Optimization (HIO) strategy **significantly enhances the generalizability of hallucination mitigation**. Specifically, HIO induces multiple potential hallucinations (as claimed in Sec.4.2) for the current token using a beam search mechanism based on their predictive logits. Given that these potential hallucinations can be any word within a 32,000-word dictionary in the LLM, HIO theoretically has the capacity to generalize to unseen objects, effectively mitigating hallucinations.
> Futhermoe, to validate the generalizability of our method, we collected samples containing out-of-distribution object categories and evaluated the model's performance on these unseen classes. The test sets, Unseen-N and Unseen-P, consist of 426 images crawled from the Internet and 69 images sourced from MSCOCO, A-OKVQA, and GQA, with no overlap with the model's training set. These test sets include a total of 10 unseen categories. The evaluation results, shown in Table 1, demonstrate that our method outperforms the baseline model (i.e., Regular), highlighting its superior generalizability to open-vocabulary objects.
>
> Table 1: Generalization experiments of our method on two unseen datasets
> |  Dataset  | Decoding | Accuracy↑ | Precision | Recall | F1 Score↑ |
> |  ---- | ---- | ----  | ----  | ---- | ----  |
> | unseen-N   |  Regular baseline | 88.88 | 84.88 | **95.63** | 83.93 |
> | |   Ours  | **92.97** | **91.94** | 94.75 | **93.33** |
> | unseen-P  |  Regular baseline | 81.15 | 64.86 | **100.00** | 78.68 |
> |   |   Ours   | **85.51** | **75.01** | 87.51 |  **80.76** |
>
> #### W2: Authors are suggested to include some results comparing these methods. Whether an amateur model can help an RLHF model seems an interesting question and may improve this paper.
> [A] Aligning Large Multimodal Models with Factually Augmented RLHF.
>
> **W2-A2:**
> Following your suggestion, we have included experimental comparisons with method [A] under the POPE benchmark. As shown in Table 2, **our method consistently outperforms LLaVA-RLHF7B across three settings in POPE**, demonstrating the effectiveness of our approach in mitigating hallucinations. Furthermore, we believe **our method can help RLHF models generate multiple hallucinated answers with minimal effort**, thereby augmenting method [A] with high-quality instruction-tuning data for fine-grained RLHF alignment. We plan to explore this in future work.
>
> Table 2:Comparative experiments between our method and the RLHF method on POPE
> |  Model  | Random |   |  |Popular  |   | | Adversarial  |   | | Overall  |   |
> |  ----   | ----   | ----   | ----   | ----  | ----  | ----  | ----  | ----   |----   |----   |----   |
> |  - | Acc↑  |  F1↑ | Yes (%)  | Acc↑ |  F1↑ |  Yes (%)  | Acc↑ |  F1↑ |  Yes (%) |  F1↑ |  Yes (%) |
> | Shikra  | 86.9  | 86.2  | 43.3 |  84.0  | 83.2  | 45.2 |  83.1 |  82.5  | 46.5 |  84.0 |  45.0 |
> | mPLUG-Owl7B |  54.0  | 68.4 |  95.6 |  50.9 |  66.9 |  98.6  | 50.7 |  66.8 |  98.7 |  67.2  | 97.6|
> | LLaVA-SFT+7B | 86.1 |  85.5 |  44.5 |  82.9 |  82.4 |  47.2 |  80.2 |  80.1 |  49.6 |  82.7 |  47.1 |
> | LLaVA-RLHF7B  | 84.8 | 83.3 |  39.6 |  83.3  | 81.8  | 41.8  | 80.7  | 79.5  | 44.0  | 81.5 |  41.8 |
> | LLaVA+Ours7B  | **90.2** | **89.9** |  58.2 |  **88.1**  | **86.8**  | 48.7  | **84.3**  | **84.3**  | 51.5  | **87.5** |  52.8 |
>
>
> #### Q1:Authors are suggested to include the other two as well for more comprehensive evaluations.
> **Q1-A3:** In response to your suggestion, **we conducted additional experiments on the MSCOCO, GQA, and A-OKVQA datasets for a more comprehensive evaluation**. The results, presented in Table 3, show that our method consistently outperforms VCD across all three setups, demonstrating its effectiveness in mitigating hallucinations in LVLMs.
>
> Table 3: Results of our method on MSCOCO, A-OKVQA, and GQA
> |  Dataset   | Setting  | Decoding | Accuracy↑ | Precision | Recall | F1 Score↑ |
> |  ----  | ----  | ----  | ----  | ----  | ----  | ----  |
> | MSCOCO  |Random    |  Regular | 83.29 | 92.13 | 72.80 | 81.33 |
> |  | |   VCD     | 87.73 | 91.42 | 83.28 | 87.16 |
> |   |  |   Ours     | **90.21** | **93.23** | **86.85** | **89.94** |
> |  |Popular      |  Regular | 81.88 | 88.93 | 72.80 | 80.06 |
> |  |    |   VCD     | 85.38 | 86.92 | 83.28 | 85.06 |
> |   |    |   Ours     | **88.13** | **88.96** | **86.83** | **87.84** |
> |   |Adversarial|  Regular | 78.96 | 83.06 | 72.75 | 77.57 |
> |  |   |   VCD     | 80.88 | 79.45 | 83.29 | 81.33 |
> | |   |   Ours     | **84.32** | **84.28** | **84.33** | **84.34** |
> | A-OKVQA  |Random |  Regular | 83.45 | 87.24 | 78.36 | 82.56 |
> |   |  |   VCD  | 86.15 | 85.18 | **87.53** | 86.34 |
> | |   |   Ours   | **90.61** | **94.97** | 85.73 | **90.19** |
> | |Popular    |  Regular | 79.90 | 80.85 | 78.36 | 79.59 |
> |||   VCD   | 81.85 | 78.60 | **87.5**3 | 82.82 |
> |||   Ours     | **86.93** | **87.84** | 85.73 | **86.77** |
> ||Adversarial|  Regular | 74.04 | 72.08 | **78.49** | 75.15 |
> |||VCD | 74.97 | 70.01 | **87.36** | 77.73 |
> |||Ours | **80.83** | 78.08 | 85.73 |**82.71** |
> |GQA|Random    |  Regular | 83.73 | 87.16 | 79.12 | 82.95 |
> |||VCD| 86.65 | 84.85 | **89.24** | 86.99|
> |||Ours| **89.06** | **93.53** | 83.93 | **88.47**|
> ||Popular |  Regular | 78.17| 77.64| 79.12| 78.37|
> |||VCD | 80.73| 76.26| **89.24**| 82.24|
> |||Ours| **84.76**| **85.35** | 83.93| **84.63**|
> ||Adversarial|  Regular | 75.08| 73.19| 79.16 76.06|
> |||VCD| 76.09| 70.83 | **88.75**| 78.78|
> |||Ours| **82.11** | **80.96**| 83.93| **82.42**|
>
> #### Q2: Typos. In line 135.
> **Q2-A4:** Thank you for highlighting the issue. We have addressed it in our revised manuscript.

---

> > ### Comment · Reviewer_6ngE · 2024-08-11
> >
> > Thanks for authors detailed reply and more evaluation experiments that address my concerns. I tend to accept this paper thus keep my earlier rating.

---

### Official Review · Reviewer_pE7J · 2024-07-10

**Soundness:** 3
**Presentation:** 3
**Contribution:** 3
**Rating:** 7
**Confidence:** 4

**Summary:**

This manuscript presents a novel perspective on implementing contrastive decoding for hallucination mitigation in large visual language models. In contrast to methods such as image perturbation, it achieves a quasi-min-max optimization by enhancing model hallucination followed by contrastive decoding for multiple hallucination targets. The proposed framework and its improved components, i.e., CBTM, AMTH, and ACI, have solid theoretical and empirical derivations. The overall writing of the paper is commendable, and it has achieved state-of-the-art performance.

**Strengths:**

The authors provide a fresh perspective on the overall uncertainty in visual input as well as the exploitation of multiple hallucinatory targets, demonstrating notable gains. They observe and deduce the limitations of the existing optimization frameworks and offer effective solutions.

**Weaknesses:**

1. The description of experimental details is weak. For instance, the experimental setups for Figure 1, Table 2, and Table 5 are not clearly stated, yet the results vary significantly.
2. The manuscript fails to provide the results of the three models purportedly studied.
3. The training details, time and hardware expenditure, and the selection of the alpha parameter, among many other experimental details for the Evil Model, have not been provided.
4. While the overall writing is satisfactory, many details betray a rather hasty preparation of the article, such as the repeated occurrences of acronym definitions, entirely duplicated result descriptions, missing spaces before parentheses in several instances, inconsistent terminologies used for metrics, etc.
5. There appears to be an error in Figure 2: 'benches' should be the correct category?
6. Inappropriate caption for Table 4.
7. Inappropriate bolding for Tables 1 and 3.
8. A Conclusion section is missing from the main text.

**Questions:**

1. In analogy to adversarial training, the utilization of hallucinatory information during the training dynamics is unclear. The subsection 'Acquisition of Multiple Candidate Hallucinations' seems relevant to this aspect, yet the discussion could be further enriched.
2. In comparison to the proposed method, is it feasible to directly calibrate the hallucinatory and targeted logits and achieve both efficiency and effectiveness?
3. Please carefully revise the writing details and enrich the experimental descriptions. I believe this will significantly enhance the clarity and soundness of the paper.

**Limitations:**

The authors have noted the additional overhead introduced by the proposed method during model training and inference. The potential social impact is positive, yet the discussion on this is lacking. The reviewer thinks that the current limitations in relevant datasets may also constrain the method's development. How to seek the annotation of hallucination tokens during dynamic optimization merits consideration.

---

> ### Author Rebuttal · Authors · 2024-08-07
>
> #### W1: The description of experimental details is weak.
> **W1-A1:** Based on your suggestions, **we have revised the descriptions of the experiments in Tab.1, Tab.2, and Tab.5 as follows**:
> Table 1 presents the experimental results on the POPE dataset across random, popular, and adversarial settings. Our method consistently outperforms state-of-the-art decoding methods across three settings, demonstrating its effectiveness in mitigating object hallucinations in diverse scenarios.
> Tables 2 and 5: We assess our model's performance in open-ended caption generation by evaluating 500 images from COCO val2017 and val2014 in Tab.2 and Tab.5. We observed that our method significantly reduces object hallucinations in generated captions with lower CHAIRS and CHAIRI scores, while enhancing caption detail, as indicated by higher Recall scores. It demonstrates that our method can achieve an effective balance between accuracy and detail in open-ended caption generation by widening the gap between hallucinated and correct tokens.
>
> #### W2: Lack results of the three models.
> **W2-A2:** We have **included the results for the LLava, MiniGPT-4, and InstructBLIP models on the MSCOCO, A-OKVQA, and GQA datasets in Tab.1**.
>
> Table 1:Results of the three models on MSCOCO, A-OKVQA, and GQA
> |Dataset| Setting  | Decoding | Accuracy↑ | Precision | Recall | F1 Score↑ |
> |-|-|-|-|-|-|-|
> |MSCOCO|LLaVA1.5|Regular|83.2| 92.1|72.8|81.3|
> ||| VCD |87.7|91.4|83.2|87.1|
> |||Ours |**90.2* |  **93.23** |**86.8**|**89.9**|
> ||miniGPT4|Regular|67.1|69.1|66.5|67.7|
> |||VCD|69.6|72.7|66.7|69.6|
> |||Ours|**77.9**|**74.1**|**85.8**|**79.5**|
> ||InstructBLIP|Regular|80.7|81.6|79.1|80.4|
> ||| VCD|84.5|88.5|**79.3**|83.6|
> |||Ours |**87.3**| **96.1**|77.7|**85.9**|
> |A-OKVQA|LLaVA1.5|Regular|83.4|87.2|78.3|82.5|
> |||VCD|86.1|85.1|**87.5**|86.3|
> |||Ours|**90.6**|**94.9**|85.7|**90.1**|
> || miniGPT4|Regular|64.7|65.2|65.7|65.5|
> |||VCD|66.6|66.4|68.2|67.3|
> |||Ours|**74.7**|**69.4**|**88.1**|**77.6**|
> ||InstructBLIP| Regular|80.9|77.9|86.1|81.8|
> |||VCD|84.1|82.2|**87.0**|84.5|
> |||Ours|**88.5**|**90.2**|86.4|**88.3**|
> |GQA|LLaVA1.5|Regular|83.7|87.1|79.1|82.9|
> |||VCD|86.6|84.8|**89.2**|86.9|
> |||Ours| **89.0**|**93.5**|83.9|**88.4**|
> ||miniGPT4|Regular |65.1|65.3|66.7|66.0|
> |||VCD|67.0|68.3|69.0|68.6|
> |||Ours|**73.8**|**70.0**|**83.2**|**76.0**|
> ||InstructBLIP|Regular|79.6|77.1|84.2| 80.5|
> |||VCD|83.6|81.8|**86.6**|84.1|
> |||Ours|**87.2** |**89.0** |84.9| **86.9**|
>
> #### W3: Lack detailed implementation.
> **W3-A3:**  **We provide an detailed description of our implementation as follows**:
> Implementation Description: We evaluate our HIO by incorporating it into three LVLMs: LLaVA 1.5, InstructBLIP, and MiniGPT-4. For decoding, we use Llama-7B and Vicuna-7B as the linguistic decoder for LLaVA and InstructBLIP/MiniGPT-4. To ensure a fair and rigorous comparison, we adhere to the configurations and guidelines from the original works and codebases of the compared models. The training is conducted on 4x RTX 3090 GPUs for LLaVA 1.5, 8x V100 GPUs for MiniGPT-4, and 4x A6000 GPUs for InstructBLIP. Each training session lasts approximately 2-4 hours. Hyperparameters including alpha and beta are set to 1.0 and 0.1, in accordance with the VCD model's specifications.
>
>
> #### W4: Details betray a rather hasty preparation of the article.
> **W4-A4:** We have revised all your concern based on your suggestions.
>
> #### W5: Error in Figure 2.
> **W5-A5:** We have fixed this issue in our revised manscript.
>
> #### W6: Inappropriate caption for Table 4.
> **W6-A6:** Based on your suggestion, we have revised the caption for Table 4.
> **Table.4: Ablation study with different components of our model on CHAIR-COCO.**
>
> #### W7: Inappropriate bolding for Tables 1 and 3.
> **W7-A7:** We have reviewed the formatting and make sure that the bolding is applied consistently and appropriately.
>
> #### W8: Conclusion section is missing.
> **W8-A8:** Due to the length of the initial submission, the conclusion was placed in the supplementary materials. In the revised manuscript, we have included the conclusion in the main text.
>
> #### Q1: Utilization of hallucinatory information is unclear.
> **Q1-A9:** The following discussion has been incorporated into our revised manuscript:
> We **utilize hallucinatory information—multiple potential hallucinated tokens—to sharpen the contrast between hallucinated and correct tokens, thereby enhancing the effectiveness of contrastive decoding**. The rationale is that while the **single-target DPO mechanism addresses a specific hallucinated token, it may inadvertently trigger other potential hallucinations**. For example, as shown in the lower half of Fig. 2, single-target DPO can correct the hallucination of "Dogs" but may introduce "Tracks" as a new hallucination with the second-highest confidence. To strengthen the contrastive decoding process, we generate multiple potential hallucinated tokens and ensure that the model considers them simultaneously. We then apply the HIO mechanism for adversarial training between the correct token and the generated hallucinations, creating an "evil" model that favors these hallucinated tokens. **This approach significantly improves the effectiveness of contrastive decoding, as the contrastive logits of the correct token surpass those of all hallucinated tokens, a fact theoretically validated in Sec. 5**.
>
> #### Q2: Efficiency and effectiveness.
> **Q2-A10:** Unfortunately, **achieving a trade-off between efficiency and effectiveness remains challenging**. As noted in [1], using pretrained models with lower capacities (e.g., BLIP2-7B) can significantly reduce training costs but results in decreased effectiveness in mitigating hallucinations.
> [1] Contrastive Decoding: Open-ended Text Generation as Optimization
>
> #### Q3: Revise the writing details and experimental descriptions.
> **Q3-A11:** We have revised writing details and enriched experimental descriptions in Tab.1, Tab.2 and Tab.5 of Sec.6.

---

> > ### Comment · Reviewer_pE7J · 2024-08-14
> >
> > I think the manuscript has reached the standard for publication after revision, and I intend to raise my score to 7.

---

### Official Review · Reviewer_RdjK · 2024-07-12

**Soundness:** 3
**Presentation:** 3
**Contribution:** 2
**Rating:** 4
**Confidence:** 5

**Summary:**

This paper focuses on alleviating hallucinations in Large Vision-Language Models. Specifically, the authors introduce a novel optimization strategy named Hallucination-Induced Optimization (HIO). This method amplifies the contrast between hallucinatory and targeted tokens relying on a fine-tuned preference model. Finally, the extensive experimental results verify the effectiveness of HIO, which outperforms state-of-the-art methods across various benchmarks.

**Strengths:**

1. The motivation of this paper is reasonable.
2. The experimental results are significant compared to previous works.

**Weaknesses:**

1. I mainly doubt the novelty. The motivation of this paper is very similar to [1]. Please describe the difference between this work and [1].
2. This paper is not complete, lacks a Conclusion section, and the ablation study section is too short. Moreover, the implementation details are missing in the paper. Thus, I think the writing of this paper is not ready.

[1] Alleviating Hallucinations of Large Language Models through Induced Hallucinations.

**Questions:**

Please see the weaknesses.

---

> ### Author Rebuttal · Authors · 2024-08-07
>
> #### W1: Novelty: motivation is similar to [1]. [1] Alleviating Hallucinations of Large Language Models through Induced Hallucinations
> **W1-A1**: We indeed adopt the same concept of induced hallucination from ICD[1], **but we also identified two key issues of it, i.e., less effective and poor generalizability**.  To address these, we propose our Hallucination-Induced Optimization including three conponents, i.e.,CBTM, AMTH and ACI.
>
> **1.Less Effectiveness：**
>
> **(1)Sentence-Level vs. Token-Level Hallucination Induction**: The Supervised Fine-Tuning (SFT) strategy operates at the sentence level, which **limits its ability to precisely induce specific hallucinated tokens at token-level**, as discussed in Lines 203-207 of Section 4.3. This limitation impairs ICD's ability to effectively amplify the distinction between hallucinated tokens and correct tokens, thus reducing the effectiveness of Contrastive Decoding for hallucination mitigation. To address this issue, we propose a DPO-based hallucination mitigation method, **the CBTM, which precisely induces token-level hallucinations for the desired tokens**.
>
> **(2) Predefined vs. Self-Generated Hallucinations**: ICD [1] relies on hallucination samples generated by ChatGPT, rather than the model's own hallucinations as proposed in our Amplification of Multiple Targeted Hallucinations (AMTH) method (Section 4.2). Since these **externally-generated hallucinations may not accurately reflect the model's actual hallucination states, they can lead to less effective optimization or even result in overfitting**. To address this issue, **AMTH induces the model's own hallucinations, providing a more precise representation of the model's internal states**. As a result, our method more effectively targets and reduces the model's hallucinations.
> **To compare the effectiveness of our Hallucination-Induced Optimization (HIO) with ICD [1], we conduct experiments to assess their performance in mitigating open-ended and discriminative hallucinations**, as detailed in Tables 1 (CHAIR) and 2 (POPE). To implement ICD based on LLaVA, we first fine-tune an Evil-LLaVA-7B model using hallucinated answers generated by ChatGPT. We then apply this fine-tuned model to perform contrastive decoding against the original LLaVA-7B model. As shown in Tables 1 and 2, our method achieves significant improvements over ICD, demonstrating its superior capability in reducing hallucinations. **This enhancement is attributed to HIO's token-level induction, which effectively utilizes hallucinations generated by the model itself**.
>
> Table 1: Comparison with ICD on CHAIR
> |Row|Method|Length|CHAIRS↓|CHAIRI↓|Recall↑|
> |-|-|-|-|-|-|
> |1|ICD|106.3|50.8|15.0|78.5|
> |2|Ours|110.3|**41.4**|**10.5**|77.4|
>
> Table 2: Comparison with ICD on POPE
> |  Dataset   | Setting  | Decoding | Accuracy↑ | Precision | Recall | F1 Score↑ |
> |-|-|-|-|-|-|-|
> | MSCOCO |Random|ICD|89.5|88.7|90.6|89.6|
> |||Ours|**90.2**|**93.2**|86.8|**89.9**|
> ||Popular|ICD|86.1|83.1|90.6|86.7|
> |||Ours|**88.1**|**88.9**|86.8|**87.8**|
> ||Adversarial|ICD|79.7|74.3|90.6|81.7|
> |||Ours|**84.3**|**84.2**|84.3|**84.3** |
>
> **2.Poor Generability：**
> The model tends to produce numerours different hallucinations for each visual semantic scene. However, **ICD only considers only one of them annotated by ChatGPT, ignoring other possible hallucinations**. As illustrated in the lower part of Figure 2, where 'People,' 'TV,' and 'Clock' are all hallucinations. For ICD, it can only induce one hallucinated token, such as 'Dog', while ignoring the other potentional hallucinations, such as 'Clock'. Finally, ICD overfis on the annotated hallucinations given by ChatGPT, failing to address most potential hallucinations, and further reducing the model's ability to generalize to unseen hallucinations.
> In contrast, our proposed Amplification of Multiple Targeted Hallucinations (AMTH) generates multiple potential hallucinations using a beam search mechanism based on their predictive logits, as described in Section 4.2. Since these potential hallucinations can be any word from the 32,000-word dictionary of the LLM, **AMTH demonstrates improved generalizability to unseen objects, thereby more effectively mitigating hallucinations**.
> **To demonstrate the generalization capabilities of our method, we collected two OOD test set as Unseen-N and Unseen-P that the model had not encountered during training for evaluation**.
> Unseen-N contains 426 samples gathered from the web, while Unseen-P includes 69 samples collected from MSCOCO, A-OKVQA, and GQA. This resulted in a total amounts of 495 samples with 10 unseen object classes.
> From Tab.3, we observe that our method significantly outperforms ICD[1] across Accuracy, Precision and F1-Score on unseen catagories by a large margin, thereby **proving more powerful generalization capability of our proposed HIO method in comparision to ICD**.
>
> Table 3: Comparison with ICD on unseen datasets.
> |Dataset|Decoding|Accuracy↑|Precision|Recall|F1 Score↑|
> |-|-|-|-|-|-|
> |unseen-N|ICD|88.6|84.5|95.6|89.7|
> ||Ours|**92.9**|**91.9**|94.75|**93.3**|
> |unseen-P|ICD|79.7|63.1|100.0|77.4|
> ||Ours|**85.5**|**75.0**|87.5|**80.7**|
>
> #### W2: This paper is not complete.
> **W2-A2:** Due to space constraints, the conclusion section was initially included in the supplementary materials. We have incorporated it into the main manuscript in our revised submission. Moreover, **we have enriched the ablation study to analyze the generalization capability of our proposed components to unseen categories, as detailed in Tab.4**. Finally, **we have integrated the ablation study into the experimental results section, rather than presenting it separately**.
>
> Table 4: Ablation study on the generalization of each component on unseen datasets
> |Dataset|CBTM|AMTH|ACI|Accuracy↑|Precision|Recall|F1 Score↑ |
> |-|-|-|-|-|-|-|-|
> |unseen-P||||81.1|64.8|100.0|78.6|
> ||✓|||82.6|66.6| **100.0**|80.0|
> ||✓|✓||84.0|72.4| 87.5|79.2|
> ||✓|✓|✓|**85.5**|**75.0**|87.5|**80.7**|

---

### Official Review · Reviewer_DdM1 · 2024-07-13

**Soundness:** 2
**Presentation:** 1
**Contribution:** 2
**Rating:** 6
**Confidence:** 4

**Summary:**

This paper proposes a method for mitigating hallucinations by training an "evil" LLM to provide logits for contrastive decoding. This "evil" LLM is trained with a dataset that prefers hallucinated samples over true ones during fine-tuning. The logits from this "evil" LLM are then used for contrastive decoding. Experimental results validate the effectiveness of this approach.

**Strengths:**

1. The high-level idea is easy to understand.
2. The method itself is interesting and novel.

**Weaknesses:**

1. The writing is sometimes confusing and makes it difficult to understand details. For example, Sec 4.1 and 4.2  use Eqn 17 in Sec 5 as the motivation of how they design the current method and refer to it frequently. However, Eqn 17 is not explained and introduced previously, making it very difficult for the reader to understand why you design this method like this. It would be much better to explain the high level idea of eqn 17 first.
2. Besides, the figure 1 itself is also confusing. The picture provided is vague and even a human finds it difficult to identify the people and output the true answer.
3. When reading the Sec 5, it is also very confusing. How do you define the ideal logits for contras decoding? Based on my understanding, it should not equal to the hallucinatory token's logits. However, at the later part line 243, you are referring to it as the hallucinatory tokens. Besides, how do you interpret the second line of Eqn 17? I interpret J as the average logit difference between the hallucinatory token and the true token. However, I find it difficult to interpret the left part of the second line of Eq 17. What is the logic behind line 242? The whole explanation does not make sense to me.
4. For Eqn 14, is it too strong to require the minimal logit of correct tokens to be larger than the maximum logit of hallucinatory tokens? For greedy docoding, if one of the correct tokens after contrast is larger than all the hallucinatory tokens, then it will be output?

**Questions:**

Sea the weakness part.
1. Typo:
- line 127 should be "probability" instead of "probably".
- line 233 it should be $\delta^{*\\{v,x,y_{<t}\\}}$ instead of $\delta^{\prime\\{v,x,y_{<t}\\}}$

**Limitations:**

Yes, the author has pointed out the computation cost for this method.

---

> ### Author Rebuttal · Authors · 2024-08-07
>
> # Response to Reviewer DdM1
> We thank all the reviewers for the insightful comments and the recognition of our work
> **"Novel Idea"** (Reviewers #1, #3, #4)
> **"Promising Results"**(Reviewers #2, $3, #4)
> **"Interesting"**(Reviewers #1, #4)
> **"Easy to understand"**(Reviewers #1, #4). Now, we carefully answer your question as follows.
>
> #### W1: The writing is sometimes confusing and makes it difficult to understand details. For example, Sec 4.1 and 4.2 use Eqn 17 in Sec 5 as the motivation of how they design the current method and refer to it frequently. However, Eqn 17 is not explained and introduced previously, making it very difficult for the reader to understand why you design this method like this. It would be much better to explain the high level idea of eqn 17 first.
> **W1-A1**:Thanks for your suggestion, we have added the descriptions of Eqn.17 in line 141 as below:
> ... to induce more potential hallucinations for effective contrast decoding, we propose to amplify multiple hallucination tokens, building on the theoretical foundation presented in Eqn. 17 of Section 4.2. This theory demonstrates that **effective contrastive decoding requires a consistent difference between the logits of potential hallucinated tokens and the correct token**. And Section 4.3 introduces additional constraints ... .
>
> #### W2: Besides, the figure 1 itself is also confusing. The picture provided is vague and even a human finds it difficult to identify the people and output the true answer.
> **W2-A2**: As per your suggestion, **we have clarified the image and enhance its quality** to ensure that the details are more discernible.
>
> #### W3: When reading the Sec 5, it is also very confusing. How do you define the ideal logits for contras decoding? Based on my understanding, it should not equal to the hallucinatory token's logits. However, at the later part line 243, you are referring to it as the hallucinatory tokens. Besides, how do you interpret the second line of Eqn 17? I interpret J as the average logit difference between the hallucinatory token and the true token. However, I find it difficult to interpret the left part of the second line of Eq 17. What is the logic behind line 242? The whole explanation does not make sense to me.
> **W3-A3**:
> **(a)** The ideal logits for effective Contrastive Decoding, as we defined, **require that the logits of all hallucinated tokens should be lower than those of the correct token**. The rationale behind this is that traditional DPO-based Contrastive Decoding methods, while reducing hallucinations on the preferred hallucinated token, often generate new hallucinated tokens, thereby diminishing the effectiveness of contrastive decoding. Therefore, we assert that the model should simultaneously reduce the logits of multiple potential hallucinated tokens so that they all remain lower than the logits of the correct token, thereby significantly enhancing the effectiveness of Contrastive Decoding in mitigating hallucinations.
>
> **(b)** Eqn.17 illustrates that hallucinations can be effectively eliminated through contrastive decoding if **the difference between the logits of the hallucinatory token and the correct token in the "evil" LVLM's output (Left part of Eqn.17) exceeds that in the original LVLM output (J in Eqn.17)**. For example, as depicted in the lower part of Figure 2, where "Dogs" is a hallucination and "Benches" is the correct label, the hallucination of "Dogs" is removed when the difference between the logits for "Dogs" and "Benches" in the "evil" LVLM output surpasses the difference in the original LVLM output. When this condition is met for all potential hallucinations, all hallucinations are effectively eliminated. A more detailed explanation will be provided in line 242 of the revised manuscript.
>
>
>
> #### W4: For Eqn 14, is it too strong to require the minimal logit of correct tokens to be larger than the maximum logit of hallucinatory tokens? For greedy docoding, if one of the correct tokens after contrast is larger than all the hallucinatory tokens, then it will be output?
> **W4-A4**:
> **(a)** Eqn.14 represents a **theoretical upper bound, which guides us in enhancing the effectiveness of Contrast Decoding method for hallucination elimination** by ensuring that the logits of all hallucinated words are lower than those of the correct words. We further refine this upper bound into a more practical optimization objective, as detailed in Eq.17. Specifically, the induced Evil model must simultaneously maintain lower logits of multiple hallucinated words than that the correct token. To achieve this, we propose the Hallucination-Induced Optimization (HIO) method, which significantly improves the effectiveness of hallucination elimination in contrast decoding.
> **(b)** If the logit of the correct token after contrast is greater than all hallucinatory tokens, **it would be selected during greedy decoding**.
>
>
> #### Q1: Line 127 should be "probability" instead of "probably"
> **Q1-A5**: Thank you for highlighting this issue, we have corrected it in our revised manscript.

---

### Author Rebuttal · Authors · 2024-08-07

We thank all the reviewers for the insightful comments and the recognition of our work.

**"Novel Idea"** (Reviewers #1, #3, #4)
**"Promising Results"**(Reviewers #2, $3, #4)
**"Interesting"**(Reviewers #1, #4)
**"Easy to understand"**(Reviewers #1, #4).


**Summary of Strengths:**

**R1-S1.** The high-level idea is **easy to understand**.
**R1-S2.** The method itself is **interesting and novel**.
**R2-S1.** The motivation of this paper is **reasonable**.
**R2-S2.** The experimental results are **significant compared to previous works**.
**R3-S1.** The work provides a **fresh perspective** to exploit hallucinatory mitigation with **notable gains**.
**R4-S1.** The method is simple and **easy to follow**.
**R4-S2.** It is an **interesting** perspective to train an amateur model first using DPO approach.
**R4-S3.** Writing and presentations are **clear**.
**R4-S4.** The performance gains against previous methods **look good**.


We have checked the manuscript carefully and made **all necessary revisions** strictly following the kind suggestions. Next, we answer your concerns **point-by-point**.

---

### Decision · Program_Chairs · 2024-09-25

**Decision:**

Accept (poster)

**Comment:**

This work deals with hallucination in large visual language models, which is a common and important problem. It provides a novel perspective on the overall uncertainty of the visual input and the exploitation of multiple hallucinatory targets. The authors have observed and mitigated the limitations of existing optimisation frameworks and have provided an effective solution. Due to the strengths of the work, three reviewers recommend acceptance. Meanwhile, one reviewer has doubts about the novelty of this work, and for this, the authors have provided detailed response showing the contributions of this work. AC thus recommends acceptance.